# The genome of Aechmea fasciata provides insights into the evolution of tank epiphytic habits and ethylene-induced flowering

Zhiying Li[1,2,3,4,6], Jiabin Wang[1,2,3,4,6], Xuanbing Zhang[5], GuoPeng Zhu[5], Yunliu Fu[1,2,3,4], Yonglin Jing[1,2,3,4], Bilan huang[1,2,3,4], Xiaobing Wang[1,2,3,4], Chunyang Meng[1,2,3,4], Qingquan Yang[1,2,3,4] & Li Xu [1,2,3,4]✉

*Aechmea fasciata* is one of the most popular bromeliads and bears a water-impounding tank with a vase-like rosette. The tank habit is a key innovation that has promoted diversity among bromeliads. To reveal the genomic basis of tank habit formation and ethylene-induced flowering, we sequenced the genome of *A. fasciata* and assembled 352 Mb of sequences into 24 chromosomes. Comparative genomic analysis showed that the chromosomes experienced at least two fissions and two fusions from the ancestral genome of *A. fasciata* and *Ananas comosus*. The gibberellin receptor gene *GID1C-like* was duplicated by a segmental duplication event. This duplication may affect GA signalling and promote rosette expansion, which may permit water-impounding tank formation. During ethylene-induced flowering, *AfFTL2* expression is induced and targets the *EIN3* binding site 'ATGTAC' by *AfEIL1*-like. The data provided here will serve as an important resource for studying the evolution and mechanisms underlying flowering time regulation in bromeliads.

[1] Institute of Tropical Crop Genetic Resources, Chinese Academy of Tropical Agricultural Sciences, Danzhou 571737 Hainan, China. [2] Ministry of Agriculture Key Laboratory of Crop Gene Resources and Germplasm Enhancement in Southern China, Danzhou 571737 Hainan, China. [3] Hainan Province Key Laboratory of Tropical Crops Germplasm Resources Genetic Improvement and Innovation, Danzhou 571737 Hainan, China. [4] National Gene Bank of Tropical Crops, Danzhou 571700 Hainan, China. [5] College of Horticulture and Landscape Architecture, Hainan University, Haikou 570228, China. [6]These authors contributed equally: Zhiying Li, Jiabin Wang. ✉email: xllzy@263.net

*A*echmea fasciata, also called the urn plant or silver vase, is a common house plant and popular bromeliad. Bromeliaceae contains more than 75 genera and more than 3500 species and is one of the largest families of flowering plants found in the neotropics[1–3]. The chromosome number of bromeliads varies little, with most species of Bromelioideae, Puyoideae, Tillandsioideae and Pitcairnioideae being diploid at 2n = 50 and some Tillandsioideae being diploid at 2n = 48[4,5]. One of the most diverse clades from Bromeliaceae, core bromelioids often use CAM photosynthesis and have absorptive foliar trichomes and 'tanks' formed by closely overlapping bases of rosette leaves to impound water and detritus, allowing these plants to absorb water and nutrients on epiphytic perches and rocks and adapt to arid regions[1,6–8].

The tank habit, CAM photosynthesis, absorptive leaf trichomes, epiphytism and avian pollination are thought to be the key innovations that have allowed bromeliads to spread into the treetops of rainforests, semiarid and arid regions and microsites and become highly diversified[2,9,10]. The tank habit and CAM photosynthesis arose several times independently within the family; for Bromelioideae, CAM photosynthesis arose at the base of Bromelioideae-Puyoideae ca. 10.7 Mya in the Andes/central Chile;[11,12] later, the tank habit coincided with epiphytism, which arose in Bromelioideae ca. 5.6 Mya in the Atlantic Forest region[8]. The tank epiphytic bromelioids had the highest net diversification in Bromeliaceae because CAM photosynthesis was associated with a twofold increase in speciation rate in the tank habit that was five times lower than that in tank-less habits[9]. Whereas Bromelioideae contains 33 genera and ca. 800 species, tank-less *Ananas* contains 7 species, but tank-habit *Aechmea* contains 260 species[13]. However, the genomic basis of these functional traits is poorly understood.

Ethylene-induced flowering is a remarkable feature of Bromeliaceae species and is exploited worldwide to promote flowering synchronization[14]. The flowering synchronization is of critical importance in *Ananas comosus* var. *comosus* and other ornamental bromeliads cultivation due to the occurrence of natural flowering out of season can cause serious scheduling problems for growers. Although ethylene promotes flower induction in Bromeliaceae species, ethylene commonly delay flowering in many plant species, including rice[15], pharbitis[16] and Arabidopsis[17]. Ethylene signalling modulates *GA–DELLA* signalling pathways that delay flowering by repressing the expression of flowering integrator genes such as *FLOWERING LOCUS T* (*FT*) and *SUPPRESSOR OF CONSTANS OVEREXPRESSION 1* (*SOC1*) and the floral meristem identity gene *LFY*[18,19]. As an exceptional case, ethylene signalling activated the expression of *FT* in *Ananas comosus* var. *comosus*[20] and *A. fasciata*[21]. But the molecular mechanism underlying ethylene-induced flowering in bromeliads remains unexplored.

The publishing of the genome of the famous fruit crop *A. comosus* var. *comosus* and *A. comosus* var. *bracteatus* provides the possibility to study the genomic basis of divergence between *A. fasciata* and *A. comosus* and the rise of tank epiphytic habits[22,23]. Here, we report the sequencing of *A. fasciata*. Ancestral genome construction and comparative genome analysis show the evolution of chromosome numbers and the loss of orthologous sets. In *A. fasciata*, the gibberellin receptor gene *GID1C-like* was duplicated; along with the insertion of 14 and 27 amino acids and multiple nonsynonymous mutations in the duplicated gene pairs relative to *AcGID1C-like* due to a segmental duplication event followed by mutation, it may affect GA signalling and promote rosette expansion, which allow water-impounding tank formation. Through analysis of the evolution of the *FT* family, coupling ChIP-seq, ChIP-qPCR and Y1H, we hypothesized that *AfEIL1* can directly bind upstream of *AfFTL2*

proteins and that *AfFTL2* may be the key gene of ethylene-induced flowering. The resulting genomic information will be helpful for studying the evolution and mechanisms of flowering time regulation in bromeliads.

## Results

**Genome assembly and annotation**. K-mer analysis of the *A. fasciata* genome estimated that the genome size was 359 Mb, with 1.15% heterozygosity (Supplementary Fig. 1). To overcome the impact of its high heterozygosity, we generated 40 Gb PacBio reads, 60 Gb Nanopore reads, and 158 Gb Illumina short reads for genome assembly. Using Nextdenovo (https://github.com/Nextomics/NextDenovo) and Nextpolish[24], we constructed 528.98 Mb scaffold sequences, with an N50 of 4.68 Mb (Supplementary Table 1). After purging redundant sequences, 352 Mb sequences were used for Hi-C mapping (Supplementary Fig. 2). Then, we constructed 24 chromosomes with 347 Mb genome sequences, which was 96.66% of the estimated genome size (Fig. 1).

We processed RNA-seq data from roots, flowers, central leaves, mature leaves and central leaves under ethylene treatment to provide transcriptional evidence supporting the annotation and to obtain reliable gene structure annotation. Using the Marker2 pipeline[25], we predicted the genes of the *A. fasciata* genome. Gene prediction produced 26,126 protein-coding genes (Supplementary Table 2), 348 tRNA genes and 104 rRNAs. BUSCO[26] analysis shows that 93.4% of completed BUSCO orthologues were predicted (Supplementary Table 3). We also identified 1473 pseudogenes and predicted 1,239 transcription factors.

We used LTR-Retriver[27] and BUSCO to estimate genome completeness. The LAI value was 13.94, and 98.4% of the BUSCO orthologues were completed, suggesting a highly complete and reference quality draft genome assembly (Supplementary Table 4). The repeat sequences consisted of 61.72% of the genome (Supplementary Table 5). A total of 0.63% consisted of transposon elements (TEs), and 58.15% of the genome consisted of retrotransposons. The LTR family Gypsy accounted for 24.31% of the genome. LTRs are thought to play a major role in the evolution of chromosome structure, specifically in repeat-dense regions such as centromeres. The LTR distribution of each chromosome was analysed, showing that LTRs were enriched near the ends of most chromosomes (18) and that a few (6) were enriched in the centre of the chromosomes (Supplementary Fig. 3). The LTR insertion time was analysed using LTR_retriever, suggesting that there was a burst of LTR insertions ca. 0.91 Mya (Fig. 2a).

**Ancestral genome construction**. We constructed the ancestral genomes of *A. fasciata*, *A. comosus* var. *comosus* and *A. comosus* var. *bracteatus* based on homologous genes detected by Orthofinder[28] according to the BLAST results for proteins (Fig. 2d). The ancestral genome contains 24 ancestral chromosomes and 13,837 ordered protogenes, including 13,119 genes of *A. comosus* var. *comosus* and 12,311 genes of *A. fasciata* and 11,445 genes of *A. comosus* var. *bracteatus*. The 24 *A. fasciata* chromosomes had experienced at least two fissions and two fusions from the ancestral chromosomes, and *A. comosus* var. *comosus* and *A. comosus* var. *bracteatus* had experienced one fission and formed the current 25 chromosomes. Chr17 of the ancestral chromosomes fissioned into chr5 and chr10 in *A. comosus* var. *comosus* and into chr3 and chr11 in *A. comosus* var. *bracteatus*, while in *A. fasciata*, the ancestral chromosome chr17 fissioned into three parts, two of which formed chr5 and chr24, and the last part fused with the ancestral chromosomes chr8 and chr21 into the current chromosome chr1 (Fig. 2e, Supplementary

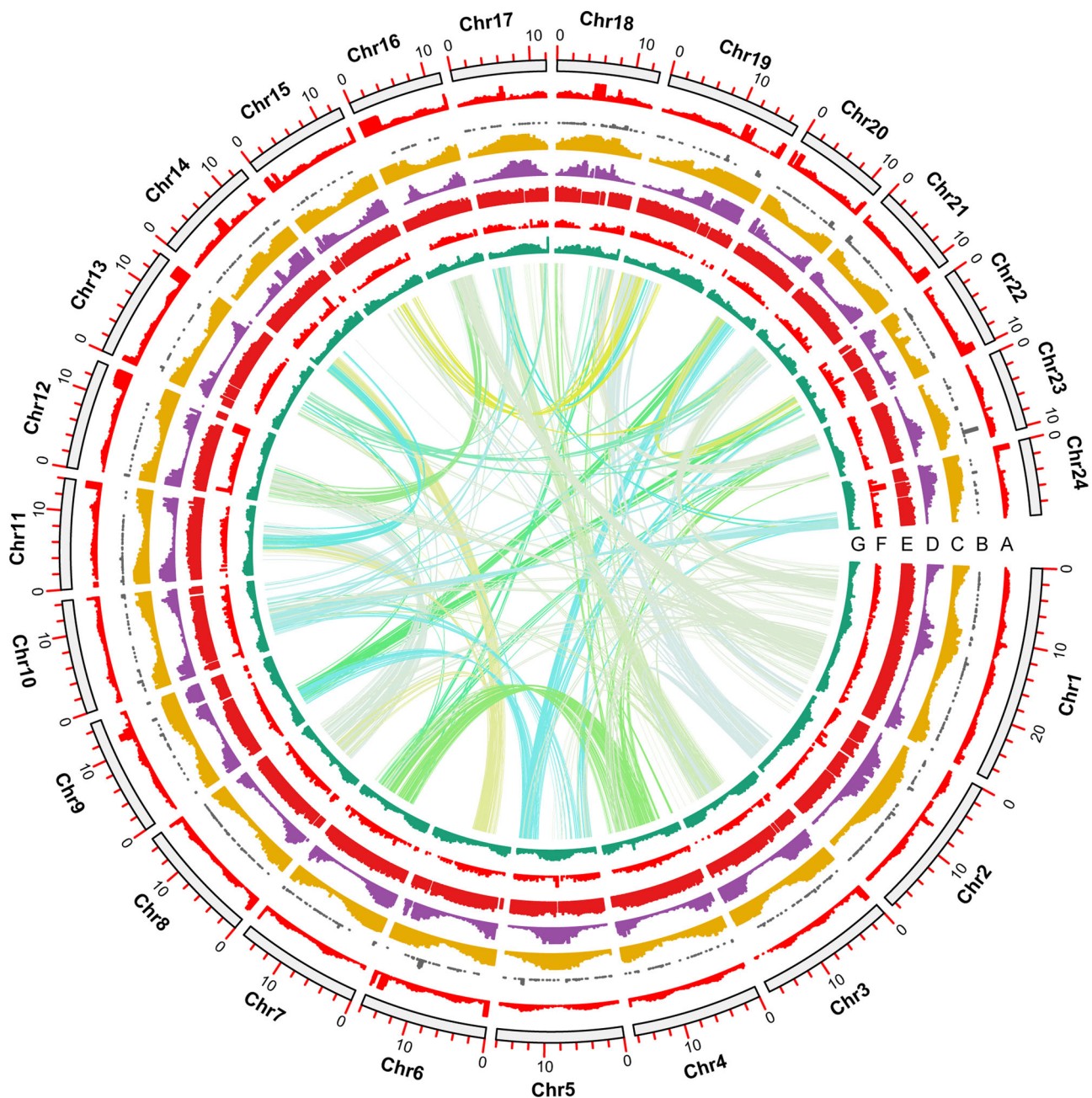

**Fig. 1 Overview of the A. fasciata genome. a** DNA Retrotransposon coverage. **b** Pseudogene distribution density. **c** Repeat sequence coverage. **d** RNA retrotransposon coverage. **e** Gene expression level. **f** Gene density. **g** GC content. The genome features are shown in 1 Mb intervals. The gene expression levels were normalized to the RPKM values.

Fig. 4 and Supplementary Fig. 5). Multiple translocations, inversions, and shifts were identified in the three genomes through synteny alignment with the ancestral genome. In particular, we identified three translocations of ancestral chr9, chr14 and chr21 into the 5' of chr15, forming the current chr23 of *A. comosus* var. *bracteatus*.

**Gene family expansion and contraction**. We identified 15,252 gene families (orthologue gene set) in *A. fasciata*, 15,999 in *A. comosus* var. *comosus* and 14,011 in *A. comosus* var. *bracteatus*, including 18,138, 20,235 and 20,699 genes, respectively. They shared 11,996 gene families, with 627, 1383 and 2618 gene families lost in *A. comosus* var. *comosus*, *A. comosus* var. *bracteatus*, and *A. fasciata*, respectively (Fig. 3b); of the gene

families, 99.21%, 98.97% and 99.36% had gene family sizes less than 5 (Fig. 3c). There were 8,092 single copy genes in the three genomes. Compared with those of *A. comosus* var. *comosus* and *A. comosus* var. *bracteatus*, the number of gene families of size more than 1 was decreased significantly in *A. fasciata*. Using DupGeneFinder[29], we predicted the duplicated gene pairs of the genome. As shown in Fig. 3a, *A. fasciata* harboured more duplicate gene pairs than *A. comosus* var. *comosus* and *A. comosus* var. *bracteatus*, with 3,112, 1,069, 914, 6757 and 23,537 whole genome duplication (WGD), tandem (TD), proximal (PD), transposed (TRD) and dispersed (DSD) gene pairs and 3062, 1072, 489, 2506 and 13,892 such pairs in *A. comosus* var. *comosus*, respectively.

We also calculated the synonymous substitutions per synonymous site (Ks) of the three genomes. There were similar WGD

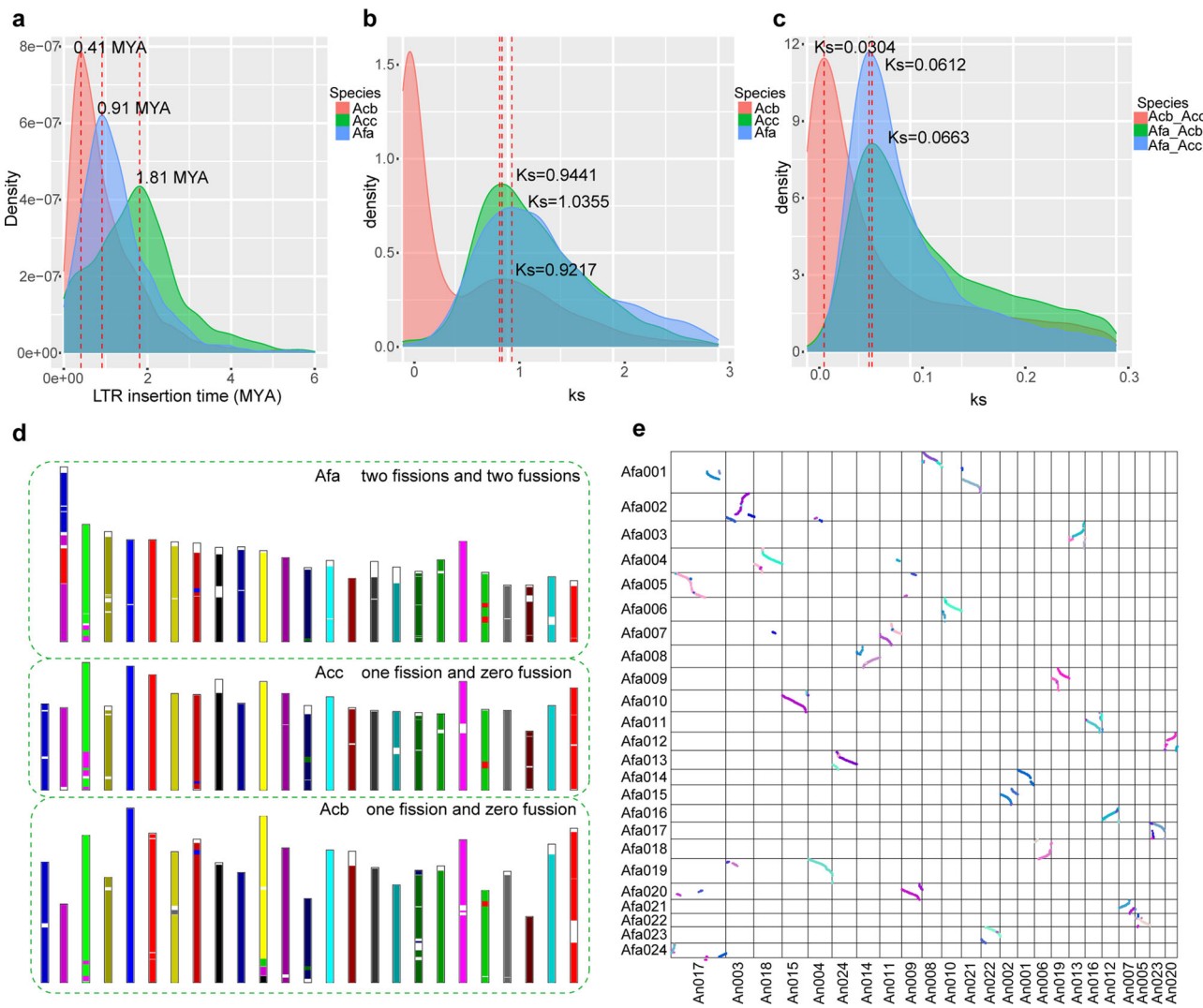

**Fig. 2 Genome evolution of *A. fasciata*, *A. comosus* var. *comosus* and *A. comosus* var. *bracteatus*. a** LTR insertion time estimation. **b** Ks distribution of Afa, Acc and Acb. **c** Ks distribution of Afa-Acc, Afa-Acb and Acb-Acc. **d** Bar plots of Afa, Acc and Acb with respect to the ancient genome. **e** Dot plot of the Afa and ancestral genomes. Afa: *A. fasciata*; Acc: *A. comosus* var. *comosus*; and Acb: *A. comosus* var. *bracteatus*.

peaks in the three genomes, corresponding to WGD event occurred 100– 120 million years ago;[23] and the divergence peaks between *A. comosus* var. *comosus* and *A. comosus* var. *bracteatus* were significantly different from the peaks between *A. fasciata* and *A. comosus* var. *comosus* (Fig. 2b, c). Using the single-copy genes of *A. fasciata*, *A. comosus* var. *comosus* and *A. comosus* var. *bracteatus* and rice, we estimated the divergence time. Analysis showed that *A. comosus* var. *comosus* and *A. fasciata* diverged ca. 5.843 MYA (Fig. 3e), when the core-Bromeliaceae appeared in the Atlantic rainfall forest[8]. According to the estimated divergence time, we used Café[30] to predict gene family expansion and contraction. There were 514, 1143, and 2733 expanded and 2090, 1067, and 3204 contracted gene families in *A. fasciata*, *A. comosus* var. *comosus* and *A. comosus* var. *bracteatus*, respectively.

**Candidate genes involved in adaptation to new environments**. Of the expanded families, several may be involved in the adaptation of *A. comosus* to new environments, including metabolism genes *Bromelain* and *transporter-like genes zinc transporter 10 (ZIP10)*-like and disease resistance proteins *Receptor kinase-like protein Xa21 (XA21)*-like, *Germin-like protein (GLP)* like and *serine/threonine-protein kinase 7 long form homologue (MAIL3)*-like genes (Fig. 3f). *GLPs* contribute to broad-spectrum diseases,

and *XA21* promotes plant innate immunity[31,32]. *MAIL3* may be involved in the regulation of root and shoot development by maintaining cell division activity[33].

The transcription factor families *RICESLEEPER2-like* and *terpene synthase 10 (TPS10)-like* were expanded in *A. fasciata* (Supplementary Fig. 6 and Supplementary Fig. 7). *RICESLEEPER2* is essential for normal plant growth and development[34]. There were 8 *RICESLEEPER2-like* loci in the ancient genome of *A. fasciata*. Compared with *A. comosus* and *A. comosus* var. *bracteatus*, two clades of *Ricesleeper2-like* were expanded in *A. fasciata* due to the dispersed duplication event after speciation with *A. comosus* var. *comosus*. *TPS10* is involved in defence against oomycete infection and indirect defence against herbivores by producing mixed signalling to attract natural enemies to plants[35,36]. The *TPS10*-like gene clusters contained 5, 7 and 12 genes in *A. comosus* var. *comosus*, *A. comosus* var. *bracteatus* and *A. fasciata*, respectively. One of the loci, a homologue of rna23024 of *A. comosus* var. *comosus*, was expanded to form a cluster with 9 genes.

**Candidate genes for tank formation**. *A. fasciata* is tank-forming core bromelioids, while *A. comosus* var. *comosus* and *A. comosus var. bracteatus* are tank-less (Fig. 3d). The gibberellin (GA)

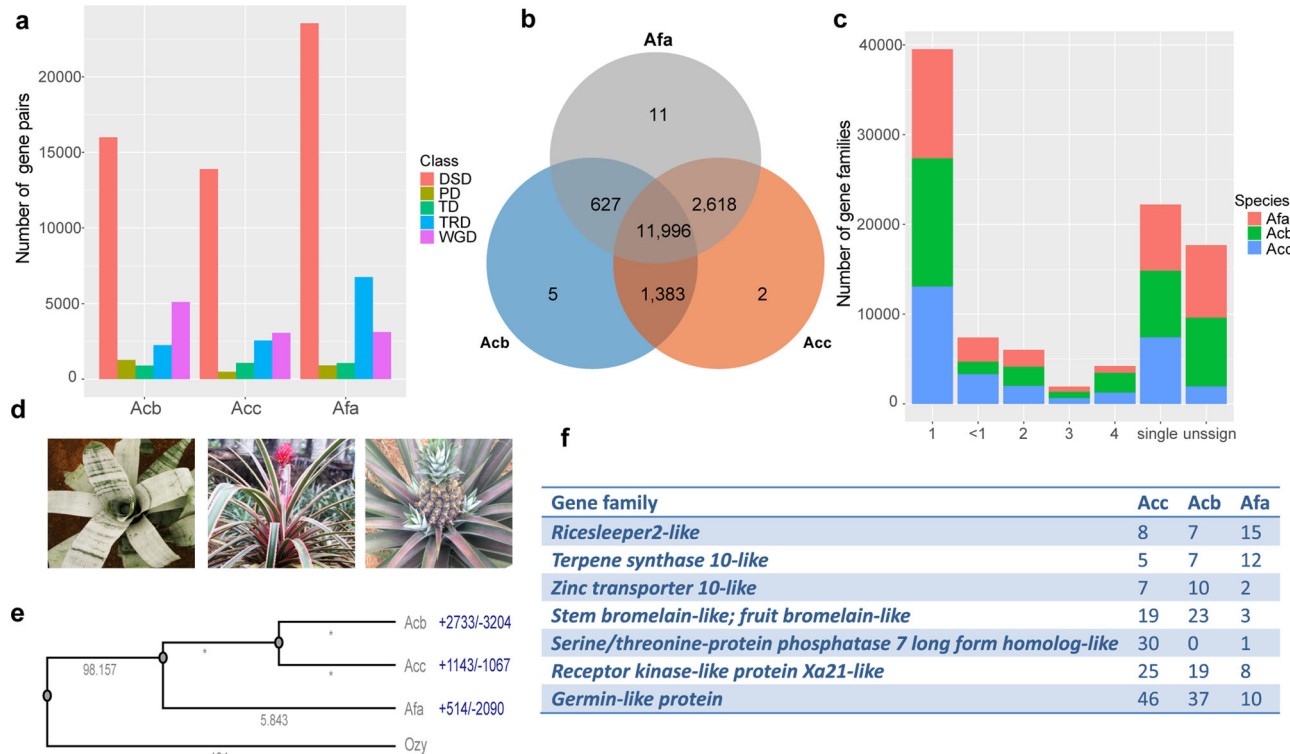

**Fig. 3 Gene family expansion and contraction estimation in *A. fasciata*, *A. comosus* var. *comosus* and *A. comosus* var. *bracteatus*. a** Duplicate gene pair distribution. **b** Venn plot of gene families. **c** Distribution of gene family size. **d** The tank-habit *A. fasciata* and tank-less pineapple and red pineapple. **e** Estimation of divergence time and gene family expansion and contraction. **f** significantly expanded or contracted gene families.

receptor gene *GID1C* was duplicated in *A. fasciata* due to segmental tandem duplication in a region of chr05, which is a conserved syntenic region with that of chr05 of *A. comosus* var. *comosus* and chr11 of *A. comosus* var. *bracteatus* (Fig. 4a). In this region, the pentatricopeptide repeat-containing proteins At2g30780-like Af03915 and Af03914 and *GID1C-like* *AfGID1CL1* (Af03916) and *AfGID1CL2* (Af03917) were tandemly duplicated gene pairs. However, the duplicated *GID1C* gene pairs had experienced different mutations in sequence. There were 14 and 27 amino acid insertions in 86–130 amino acids of *AfGID1CL2* and *AfGID1CL1*, respectively, and non-synonymous mutations at positions 10, 139, 202, 226, 229, 249, 278, 355 and 391 relative to *AcGID1C-like* (Supplementary Fig. 8 and Fig. 4b). Gene expression analysis revealed that *AfGID1CL1* and *AfGID1CL2* were expressed in all plant organs and higher in the roots, and the expression level of *AfGID1CL1* was one-fold higher than that of *AfGID1CL2* (Supplementary Fig. 9). Overexpression of *OsGID1C* resulted in a GA hypersensitivity phenotype in transgenic rice[37]. Overexpression of *GID1C-like* in Arabidopsis resulted in the expansion of rosettes[38,39]. The tandem duplication of the *GID1C-like* gene increased its expression level and may improve GA sensitivity and promote rosette expansion in *A. fasciata*, which allows the formation of tanks with closely overlapping leaves. In addition, there were 3 and 6 homologues of *DELLA* in *A. fasciata* and *A. comosus* var. *comosus*, respectively, which were distributed in conserved syntenic blocks among the chromosomes (Fig. 4c). *AfSLR1-like* and *AcSLR1-like* have complete *DELLA* and *GRAS* domains but unique motifs, such as 'LEQLD', 'MAMAMG' and 'VVHYNP' (Fig. 4d), different from the previously described 'LEQLE','-MAMGM' and 'TVHYNP' of angiosperm *DELLA*[40]. *AfSLN1-like* and *AcSLN1-like* have a 'DGLLA' motif and a unique 'LERLD' motif, which is also different from the described 'LERLE' motif of

angiosperm 'DGLLA'[40]. There is a specific 'AAAAEVEEE GEEAAEE' insertion in the *GRAS* domains of *AfSLR1-like* and *AcSLR1-like* (Supplementary Fig. 10). Moreover, four homologues lacked *DELLA* domains and partial *GRAS* domains in *A. comosus* var. *comosus*, including *AcRGL1-like*, *AcRTH1-like*, *AcSLR1-like1* and *AcSLR1-like2*. In *A. fasciata*, the homologues of *AcRGL1-like* and *AcRTH1-like* became pseudogenes, and homologues of *AcRGL1-like* were lost.

**The FT family and ethylene-induced flowering.** As shown in Fig. 5a, there are 7 *FT-like* genes in *A. fasciata*, distributed on 5 chromosomes. Compared with those in *A. comosus* var. *comosus*, the locations and gene structures of *FT-likes* in *A. fasciata* were conserved; *FTL1* and *FTL4* are tandemly duplicated genes in both *A. fasciata* and *A. comosus* var. *comosus*. The phylogenetic analysis of *FTLs* and analysis of amino acid residues showed that *AfFTL2* and *AcFTL2* are highly homologous to *Zcn8*[41,42] and *SbFT8*[43], which act as florigens in *Zea mays* and *Sorghum bicolor*, and harbour similar amide residues in segment B, which is the key amide residue for functioning as a florigen (Fig. 5b). *FTL1* shares high similarity with *HD3a* and *RFT1* but does not respond to ethylene treatment. *FTL3* is a homologue of *ZCN26*, and Segment B of the *FT* proteins varies in different *FTs*. The two lines of 35 S::AfFTL2 plants flowered early, with one of the lines flowering only with three rosette leaves (Fig. 5g). Using RNA-seq, we analysed the expression of *FT-like* genes in roots, central leaves, mature leaves, flowers and central leaves under ethylene treatment: *AfFTL4* and *AfFTL7* were not expressed; *AfFTL4* was mainly expressed in flowers; only *AfFTL6* was expressed in roots; *AfFTL2*, *AfFTL3*, *AfFTL5* and *AfFTL6* were expressed in central leaves; and only *AfFTL2* was dramatically induced by ethylene (Fig. 5c). To investigate the mechanism by which ethylene induces *AfFTL2* expression,

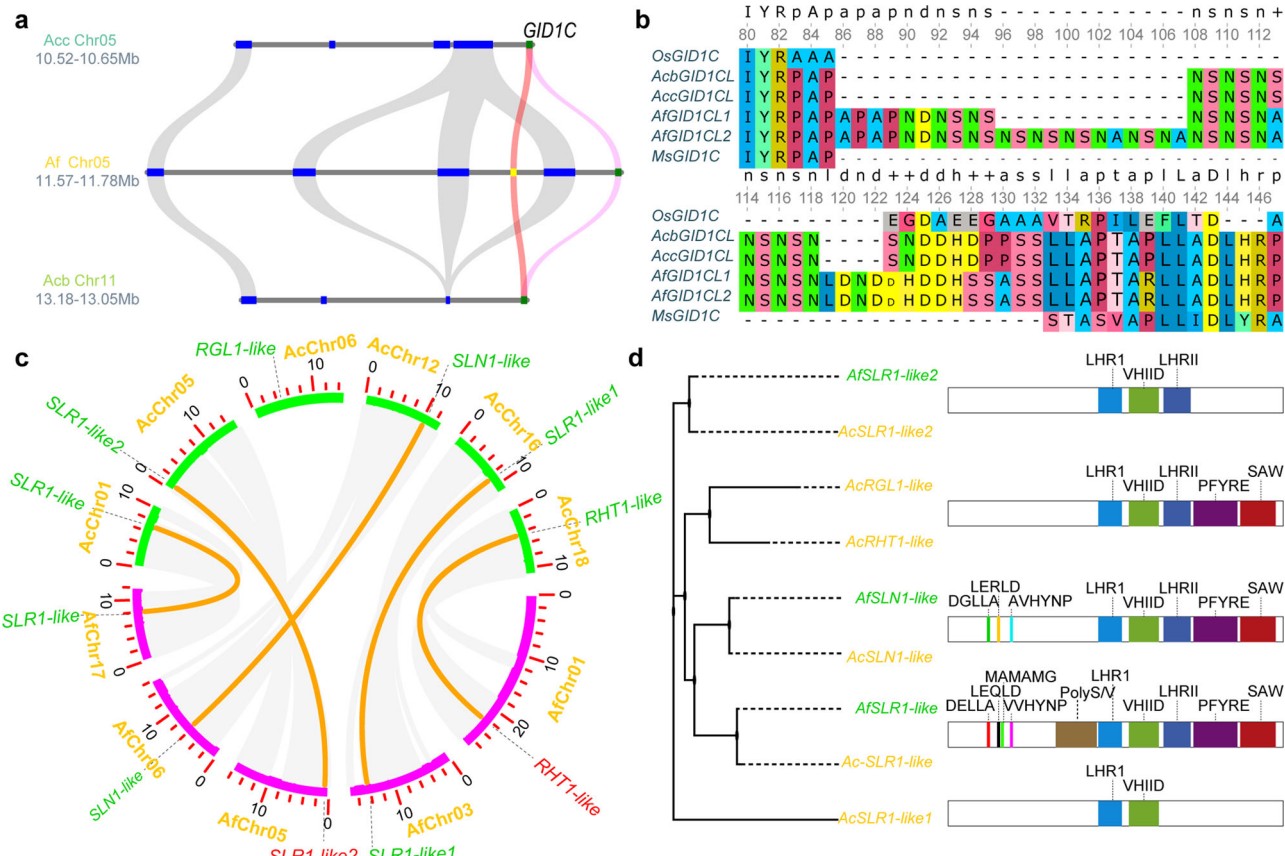

**Fig. 4 Duplication of *GID1C-like* and contraction of *DGLLA-like* in *A. fasciata*. a** Microsynteny block containing the duplicated *GID1-likes* among *A. fasciata* (Af), *A. comosus* var. *comosus* (Acc) and *A. comosus* var. *bracteatus* (Acb). **b** Portion of the multiple alignment of *GID1C-like* proteins; Os, *Oryza sativa subsp. Japonica*; Ms, *Musa acuminata subsp. malaccensis*; Acb, *A. comosus* var. *bracteatus*; Acc, *A. comosus* var. *comosus*; Af, *A. fasciata*. **c** The distribution and synteny of DELLA and DGLLA-like proteins between *A. fasciata* (Af) and *A. comosus* var. *comosus* (Acc). **d** The domain and polygenetic tree of *A. fasciata* (Af) and *A. comosus* var. *comosus* (Acc) *DELLA* and *DGLLA*-like proteins.

we conducted *AfEIL1-like* ChIP-seq and identified the core EIN3 binding motif 'ATGTAC' (Fig. 5e). ChIP-qPCR and Y1H assays validated the binding of EIN3 to *AfFTL2* 0–500 bp upstream of the promoters (Fig. 5d and Supplementary Table 6). We predicted the ethylene-related cis-elements EBS and ERE in the 2000 bp promoters of the genes (Fig. 5f). Both the *FTL3* and *FTL2* promoters harbour EBS, but *AcFTL2* does not have ERE sites, suggesting that EBS is a conserved element upstream of the *FTL2* promoters of *A. fasciata* and *A. comosus* var. *comosus*.

The *A. fasciata* homologues of five main flowering pathway genes in Arabidopsis and rice were identified by protein alignment. The flowering pathway genes were conserved in bromeliads relative to rice (Fig. 6a). The age pathway genes *AfmiR156*, *AfmiR172*, *AfTOE1* and *AfSPL14* were identified to be involved in *A. fasciata* flowering time[44,45]. However, in contrast to Arabidopsis and rice, there was an *EIL1*-dependent pathway to induce *FTL2* expression and then downstream signalling genes. In addition, ethylene signalling also induced the expression of homologues of genes involved in plant development, including *CAL-A*, *ERF3*, *ETR3*, *EXPA10*, *LEA5*, *NAKR3*, and *LSH6*, which shared similar expression with *AfFTL2* and synchronized flowering (Fig. 6b). *LSH6* plays a critical role in synchronizing flowering[46]. *NakR3* may play a role in the transport of *FT* proteins[47]. *CAL-A* functions as a floral meristem identity gene[48]. *ETR3*, *ERF03*, *OTU6*, and *EXPA10* play roles in plant growth and varied developmental processes[15,49,50].

## Discussion

Bromeliaceae arose in the Guayana Shield ca. 100 Mya[1]. The subfamily Bromelioideae is one of the most diverse clades in Bromeliaceae. The ancestral bromelioids were tank-less, coinciding with the formation of epiphytism, and the core-Bromeliodeae tank habit arose in the Atlantic Forest ca. 5.64 Mya[8]. Using rice as an outlier, we estimated that the divergence time between *A. fasciata* and rice was 104 Mya and that the divergence time between *A. fasciata* and *A. comosus* var. *comosus* was 5.843 Mya, which is close to the time of the tank-forming core-Bromeliodeae arising in the Atlantic Forest[8]. Although *A. comosus* var. *bracteatus* is a variety of pineapple, the introgression of *A. macrodontes* genes into *A. comosus* led to genetic differentiation between *A. comosus* var. *bracteatus* and *A. comosus* var. *comosus*[51,52]. This phenomenon allowed the construction of the ancestral genomes of *A. comosus*, *A. comosus* var. *bracteatus* and *A. fasciata*. Synteny alignment with the ancestral genome showed that *A. comosus* var. *bracteatus* experienced specific translocations on chr23. In addition to multiple translocations, inversions and shifts of chromosomes, *A. comosus* var. *comosus* and *A. comosus* var. *bracteatus* experienced one fusion, and *A. fasciata* experienced at least two fissions and two fusions, indicating that changes in genomic structure play key roles in tank epiphytism habit formation. Due to the chromosome fusions of ancestry chromosomes, *A. fasciata* has 48 (2n = 48) chromosomes and was validated by the root tip squash method (Supplementary Fig. 11). We also observed a karyotype with 2n = 50 due to breakage of chromosome 1 near the centromere, which

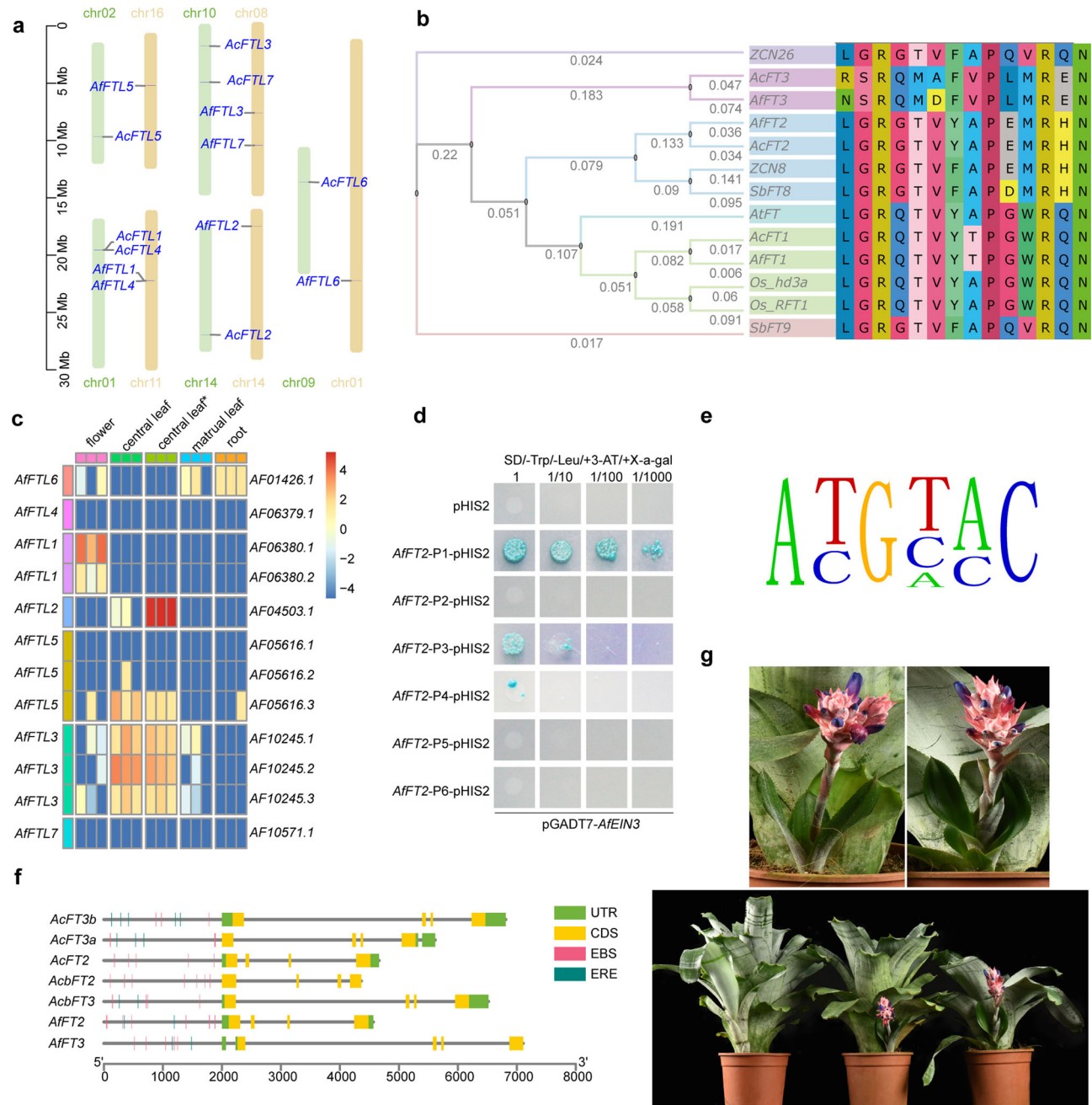

**Fig. 5 FT-like genes are the key genes involved in ethylene-induced flowering in A. fasciata. a**, FT-like distribution in A. fasciata and A. comosus var. comosus. **b** Neighbour-joining tree and key amino acid residues in segment B of the fourth exon of FT-like proteins from Os (Oryza sativa subsp. japonica), Sb (Sorghum bicolor (L.) Moench), ZCN (Zea mays), At (Arabidopsis thaliana), Af (Aechmea fasciata) and Ac (Ananas comosus). **c** Heatmap of AfFTL expression levels in flowers, roots, central leaves, and central leaves* (central leaves 24 h after ethylene treatment). **d** Yeast one-hybrid (Y1H) assay showing AfEIL1-like binding to the promoter of AfFTL2. **e** EBS motif detected by ChIP-seq in A. fasciata. **f** Gene structures of AfFTL2, AfFTL3 and their paralogous genes in Acc (Ananas comosus (L.) Merr.) and Acb (Ananas comosus var. bracteatus); the EIN3 binding sites (EBS) and ethylene response elements (ERE) were detected within 2000-bp upstream of transcription initiation sites; UTR, untranslated region; CDS coding sequence. **g** Two lines of 35 S::AfFTL2 A. fasciata plants flowered early; bottom left, wild plants; bottom centre, 35 S::AfFTL2 line 1; bottom right, 35 S::AfFTL2 line 2; upper left, flower of 35 S::AfFTL2 line 1; and upper right, flower of 35 S::AfFTL2 line 2.

may be the reason why previous reports described a karyotype with 2n = 50 chromosomes[53]. Transposon elements play key roles in the instability of the genome and genome evolution. In A. comosus var. comosus, 44% of the genome assembly consisted of TEs, and 33% consisted of LTRs, but 69% and 52% of the genome was composed of TEs and LTRs, respectively; in A. comosus var. bracteatus, 64.19% and 48.20% of the genome consisted of TEs and LTRs, respectively; and in A. fasciata, only

58.78% and 33.9% of the genome consisted of TEs and LTRs, respectively. According to the estimation of genome size, A. comosus var. comosus, A. comosus var. bracteatus and A. fasciata were 526, 591 and 359 Mb, respectively. Therefore, the main cause for the significantly smaller size of the A. fasciata genome relative to the A. comosus var. comosus and A. comosus var. bracteatus genomes was the lower insertion of TEs and LTRs. The loss of numerous genes, specifically duplicated genes, would be

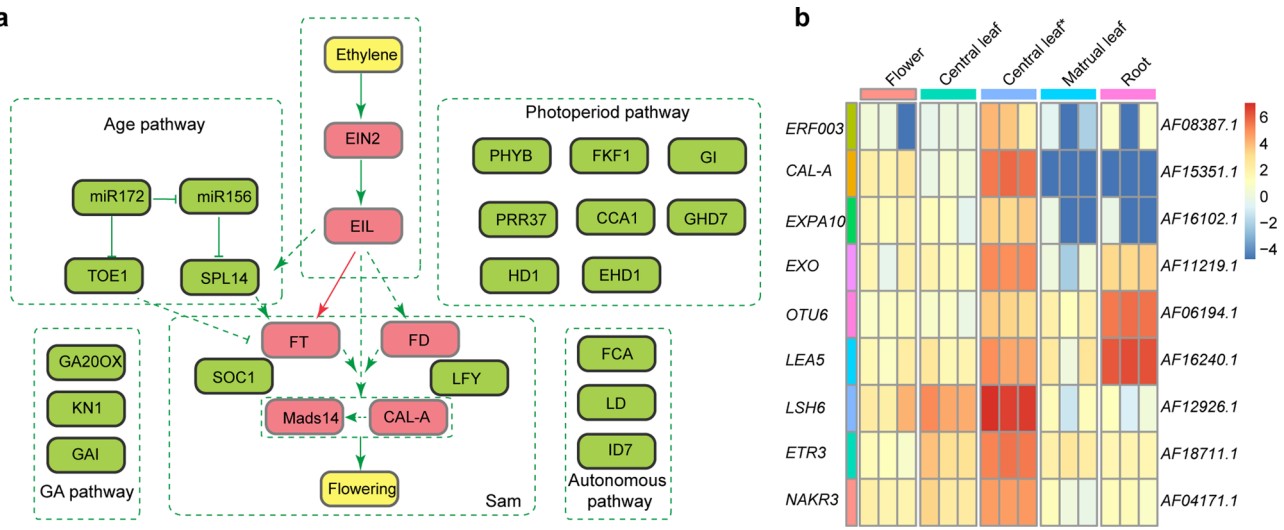

**Fig. 6 A putative flowering pathway in *A. fasciata*. a** Putative flowering pathway in A. fasciata. **b** Heatmap of putative genes synchronizing flowering in *A. fasciata*.

another reason for the significantly smaller size of the *A. fasciata* genome relative to that of *A. comosus* var. *comosus* and *A. comosus* var. *bracteatus*. A total of 1894 orthologue sets were lost in *A. fasciata*, and coupled with the loss of duplicated genes, the number of predicted protein-coding genes was 7,552 less than that of *A. comosus* var. *bracteatus*. It is supposed that WGD pairs were lost in the subsequent millions of years. The duplication of a single gene has a short half-life and occurs continuously, playing roles in key adaptations to environments[54]. According to the expansion and contraction analysis by Café[30], 1975 gene families were contracted, and only 28 gene families were expanded. The expanded gene families may play key roles in the formation of tank-epiphytic habits in A. fasciata.

With the formation of tank habits, bromeliads invaded the treetops of rainforests. Especially in Atlantic forests of Brazil, they presented with extraordinary biodiversity. The tank habit allows bromeliads to impound and absorb water and nutrients through their leaves, not requiring the uptake of water and nutrients by roots, leading to their degeneration into anchorage. Leaves of most bromeliads are organized in stemless rosettes that allow the development of aquatic and terrestrial ecosystems[55,56]. The duplication of *GID1C*-like genes and contraction of *DELLA*-like genes in *A. fasciata* may promote rosette expansion by affecting GA signalling, which allowed the rosettes to overlap closely and impound water. GA is a hormone that regulates various developmental processes, such as stem elongation, leaf expansion, flowering, seed germination and pollen development[57–59]. The effect of GA on controlling leaf expansion has been studied widely[60–64]. In the classical GA signalling pathway, GA regulates the gene expression of various pathways by interacting with the receptor *GID1* and destabilizing the *DELLA* protein, which are master regulators of the GA signalling pathway[37,65]. *DELLA* proteins have conserved eponymous DELLA motifs, followed by LEQLE, TVHYNP and MAMGM motifs in the N-terminus[40]. The GRAS domain is in the C-terminus and connects to these motifs by a poly S/T/V motif, which consists of five subdomains, including LHRI, LHRII, VHIID, PFYRE and SAW[66]. All the motifs and subdomains, except LHRI, were associated with the affinity of *DELLA* to GA preceptor *GID1*[40]. The motifs in the N-terminus function in the transactivation of plant growth regulators;[67] the motifs in the C-terminus are essential for the formation of the transactivation complex and sequestration of *DELLA*-interacting proteins; specifically, the SAW motif plays a

key role in the suppressive activity of *DELLA*[68,69]. *DELLA* functions as a master regulator in the trade-off between plant growth and various adverse environmental conditions. *AfSLR1*-like and *AcSLR1*-like proteins have unique LEQLD, MAMAMG and VVHYNP motifs in the N-terminus and AAAAEVEEE-GEEAAEE insertions in the GRAS domain, which are different from previously reported conserved motifs in *DELLA* proteins, suggesting that Af*SLR1-like* and Af*SLR1-like* may regulate different growth suppressors and *DELLA*-interacting proteins and play a pivotal role during adaptation to specific environments in bromeliads. In *A. comosus* var. *comosus* and *A. fasciata*, there were 5 and 2 *DGLLA* genes, respectively, which were DELLA-related proteins with DGLLA motifs instead of DELLA motifs[69]. *DGLLA* homologues function as back-up GA-insensitive negative regulators of plant growth under specific conditions[70]. The contraction of *DGLLA* homologues may promote the plant growth in *A. fasciata*. In conclusion, the duplication of *GID1-like* genes and contraction of *DGLLA-like* genes may play key roles in the formation of tank habits. Transposase-like genes of the HAT (hobo, activator, Tam3) family are essential for plant growth in rice and Arabidopsis[71]. *Ricesleeper*-overexpressing Arabidopsis plants showed delayed inflorescence development, increased rosette leaf number and irregularity, dwarfism and fasciation. *Ricesleeper*-overexpressing rice plants were also dwarfed[34]. Compared with *A. comosus* var. *comosus* and *A. comosus* var. *bracteatus*, *A. fasciata* contained 8 original *Ricesleeper2* loci, 2 of which had expanded. The expansion of *ricesleeper2-like* may also be associated with tank epiphytic habit formation in *A. fasciata*. In addition, Key innovations could significantly decrease the extinction rate by improving the dispersion ability, helping the plants to invade new regions or increasing defences against herbivores[9]. *TPS10* is involved in the indirect defence against lepidopteran larvae by producing a mixture that attracts natural enemies to plants being fed upon[35]. *TPS10 is* also induced by oomycete infection in *Medicago truncatula*[36]. The expansion of *AfTPS10-like* gene clusters improved the defence against herbivores and pathogens and may play a critical role in adaptation to environments. Tank-less bromeliads often inhabit semiarid or arid regions, where they lie at the limit of physiological tolerance, and most other plants fail to survive. The expansion of *AcMAIL3-*like may play a key role in the development of pineapple roots. The expansion of *ZIP10-*like may help pineapple adapt to infertile environments.

*FT* genes are highly conserved in sequence and function in angiosperms. *FT* genes regulate plant growth and multiple developmental processes, including flowering, expansion of storage organs, root development, flower development, stomatal movement and so on. In previous studies, *AFTL1* was cloned and promoted flowering in Arabidopsis. As homologues of *HD3A*, *FTL1* was expressed in flowers in both pineapple and *A. fasciata* and may play a role in flower development. *FTL3* was induced by ethylene in pineapple but not in *A. fasciata*. *FTL3 is* a homologue of *SbFT9*, is expressed in leaves and may play a role in leaf development. *AfFTL6* was expressed in the roots of *A. fasciata*, indicating that *FTL6* may be involved in root development. Of the *FTLs*, only *FTL2* was induced by ethylene and had the same amino acids in segment B as *ZCN8* and *SbFT8*, indicating that *FTL2* regulates flowering in pineapple and *A. fasciata*. Plants need to flower under optimal conditions to ensure successful reproduction. Therefore, plants have evolved a complex mechanism to integrate endogenous and environmental cues to flower at optimal times. Pineapple flowers naturally under cool night temperatures and short days[72], as does *A. fasciata*. It is thought that pineapple flowering is induced by a burst of endogenous ethylene controlled by environmental cues[73]. The ethylene synthase inhibitor amino vinylglycine could delay the flowering of pineapple[74]. Silencing *AcAcs2*, a critical ethylene biosynthesis gene, also delayed pineapple flowering[73]. In *A. fasciata*, *AfEIL1-like* could induce the expression of *AfFTL2* directly, indicating that there is an *EIL1*-dependent flowering pathway in pineapple and *A. fasciata*. Moreover, the homologues of *CAL-A*, *LSH6*, *ERF3*, *NAKR3*, etc. were induced by ethylene, similar to pineapple, indicating that there is a complex and conserved mechanism regulating flowering in bromeliads. *DELLA* functions as a flowering repressor and plays essential roles in ethylene delayed flowering by reducing bioactive GA levels and increasing *DELLA* accumulation[18]. In pineapple, ethylene treatment also reduced the bioactive GA level[75]. Ethylene-induced flowering was independent of *DELLA*-mediated flowering repression, which may be due to the unique motifs of *AfSLR1-like* and *AcSLR1-like*. Further research on the mechanism by which ethylene induces flowering is needed to facilitate plantation management and culture of new varieties.

In summary, we reported the genome sequences of *A. fasciata*, a popular ornamental tropical plant, using Nanopore, PacBio, Illumina and Hi-C sequencing to reach a high level of completeness and accuracy of the genome. The genome sequences of *A. fasciata* will facilitate further research on bromeliad evolution and the genomic basis of tank habit formation of tank bromeliads and ethylene-induced flowering.

## Methods

**Sampling, sequencing and assembly**. The *A. fasciata* plants for genome sequencing were sampled in a greenhouse of the national gene bank of tropical crops in Danzhou, Hainan, China. The genomic DNA of seedlings was extracted for genomic library construction. For Illumina sequencing, libraries with 350-bp insertions were constructed; for PacBio sequencing, 5 libraries with 20k-bp insertions were constructed and sequenced on the PacBio RS II system with P6-C4 chemistry; for Nanopore single-molecule sequencing, libraries with high molecular weight genomic DNA were constructed on PromethION. In total, 60,191,465,203 bp and 2,177,689 reads were produced by Nanopore single-molecule sequencing, with an average length of 27,640.06, a maximum length of 265,723 and an N50 of 33,856. There were 158 Gb Illumina short reads produced. There were 42,180,646,315 bp and 4,498,100 PacBio reads with an average length of 9377.4 bp and a maximum length of 53,937 bp.

Hi-C libraries were created from tender leaves of *A. fasciata* at BioMarker Technologies Company as described previously. Briefly, the leaves were fixed with formaldehyde and lysed, and then the cross-linked DNA was digested with *HindIII* overnight. Sticky ends were biotinylated and proximity-ligated to form chimeric junctions that were enriched and then physically sheared to a size of 500–700 bp. Chimeric fragments representing the original cross-linked long-distance physical interactions were then processed into paired-end sequencing libraries, and 1,001

million 150-bp paired-end reads were produced on the Illumina HiSeq X Ten platform.

For RNA-seq, total RNA was extracted from central leaves, roots, flowers and central leaves under ethylene treatment for 24 h. After removing genomic DNA using *DNase I* (Takara), mRNAs were obtained using oligo (dT) beads and broken into short fragments, followed by cDNA synthesis. Paired-end sequencing was conducted on the HiSeq X Ten platform (Illumina, CA, USA).

**Genome assembly and annotation**. Using GenomeScope[76] for k-mer analysis, the genome size of *A. fasciata* was estimated. PacBio reads were first self-corrected with the parameter corOutCoverage = 100. The corrected reads, along with NanoPore long reads, were imported for assembly by Nextdenovo v2.3.1 (https://github.com/Nextomics/NextDenovo) and NextPolish (https://github.com/Nextomics/NextPolish) using the default parameters. Then, the redundant sequences of the polished contig sequences were eliminated by Purge Haplotigs[77]. Chromosomal assembly was performed based on proximity-guided assembly using ALLHIC[78].

A de novo repeat library of the genome was customized using RepeatModeler[79], which can automatically execute two de novo repeat finding programs, RECON (version 1.08) and RepeatScout (version 1.0.5). Consensus transposable element (TE) sequences generated above were imported to RepeatMasker (version 4.05) to identify and cluster repetitive elements. Unknown TEs were further classified using TEclass (version 2.1.3). To identify tandem repeats within the genome, the Tandem Repeat Finder (TRF) package (version 4.07) was used with the modified parameters of '1 1 2 80 5 200 2,000 –d –h' to find high-order repeats.

To further investigate LTRs, we applied the LTR_retriever pipeline[27], which can integrate results from public programs such as LTR_FINDER and LTRharvest and efficiently remove false positives from the initial predictions. The predicted LTRs were further classified into intact and nonintact LTRs, and the insertion time was estimated as $T = K/2\mu$ (K is the divergence rate, and $\mu$ is the neutral mutation rate; the default is $1.38 \times 10 - 8$ in LTR_retriever) using the scripts implemented in the LTR_retriever package. Genome completeness was assessed with BUSCO v5.2.2[26].

Gene annotations were processed with MAKER2[25]. Two rounds of MAKER2 processing were used to achieve high-quality gene annotation. The RNA-seq reads were imported to Trinity, genome-guided and de novo assembled with default parameters. Then, the combined reads were imported to the PASA pipeline[80]. The PASA assembled transcripts were used to train the ab initio predictors SNAP[81], GENEMARK[82] and AUGUSTUS[83]. After that, MAKER2 processed the first gene annotation. After filtering the values of predicted proteins by MAKER2, the ab initio predictors were trained again. BRAKER[84] were used for predicting genes using aligned RNA-seq reads as input. Then, using the transcripts processed by Stringtie[85] as input and gene models predicted by BRAKER, the MAKER2 pipeline was run again. Using the GenblastA[86] pipeline and Genewise[87], we also predicted pseudogenes. PlantTFDB[88] was used for transcription factor prediction. The *A. comosus* var. *bracteatus* genomic data were downloaded from EBI-ENA (PRJEB33121)[89]. The *A. comosus* var. *comosus* genomic data were downloaded from NCBI (ASM154086v1)[90].

**Genome structure and evolution**. Orthofinder[28] was used for the identification of putative paralogous and orthologous genes from *A. fasciata* and other species. Because the coding genes have one more transcript, we retained the longest transcripts. Based on homology and collinearity, we investigated the paleohistory of *A. fasciata*, *A. comosus* var. *comosus* and *A. comosus* var. *bracteatus*. After alignment of coding genes using BLAST, we identified putative protogenes (pPGs). Then, the pPGs, ordered by location on the chromosome, were imported to MGRA2[91]. After analysis of gene gain/loss, the ancestral genome was reconstructed with an exhaustive set of ordered protogenes (oPGs). Then, the oPGs were used for collinearity analysis by MCscanX[92].

**ChIP-seq and ChIP-qPCR**. Chromatin immunoprecipitation assays were performed as previously described[93]. Briefly, 3 g of sample was washed twice in cold PBS buffer, and proteins were cross-linked to DNA by incubating the samples with formaldehyde at a final concentration of 1%. Afterwards, samples were lysed, and chromatin was obtained on ice. Chromatins were sonicated to obtain soluble sheared chromatin (average DNA length of 200–500 bp). One part of the soluble chromatin was stored at –20 °C for input DNA, and the remainder was used for immunoprecipitation by the antibodies *AfEIL1-like* and normal rabbit IgG (CST, 2729). Immunoprecipitated DNA was amplified by PCR using specific primers. All primers are listed in Supplementary Table 6.

To construct ChIP-Seq libraries, the resulting ChIP DNA above was used to generate a sequencing library according to the Illumina ChIP-Seq manufacturer's instructions. Then, an Illumina Genome Analyser II (Illumina, San Diego, CA) was used for sequencing according to the manufacturer's instructions.

**Y1H assay**. The yeast one-hybrid assay was performed as previously described[45]. The *AfEIL1-like* sequences were inserted into the pGADT7 vector (Clontech, USA) to construct pGANDT7-*AfEIL1-like* vectors. For Y1H cDNA library screening, the 200-bp promoter fragment *AfFTL2* was cloned into the destination vector pGADT7. The growth of the transformants on SD-His-Ura-Trp medium was

considered an indicator of *AfEIL1-like* binding to the corresponding DNA fragments.

**Transgenic plants**. The coding sequence of *AfFTL2* was cloned into the binary vector Cam35S under the control of the CaMV35S promoter. The confirmed construct was transformed into *A. fasciata* using agrobacterium strain GV3101 with the leaf base dip method. Transgenic plants were verified by PCR to detect exogenous genes and qRT–PCR to detect the expression of *AfFTL2*.

**Statistics and reproducibility**. Statistical analyses were performed using the s software R (version 3.6.3). The significance level was typically set to 0.05 for all the statistical analyses. The RNA-seq, ChIP-seq and ChIP-qPCR experiments were performed with three biological replicates and the merged tissues of three plants were sampled for each treatment.

**Reporting summary**. Further information on research design is available in the Nature Research Reporting Summary linked to this article.

## Data availability

The *Aechmea fasciata* genome sequences and raw sequence data from RNA-seq and genome sequencing have been deposited under BioProject accession number PRJNA748557[94]. The *AfEIL1-like* ChIP-seq data were deposited under GEO accession number GSE205799[95].

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

## Acknowledgements

This work is financially supported by the National Key Research and Development Plan Program (No. 2021YFC2600603), the National Natural Science Foundation of China (No. 31372106), the Natural Science Foundation of Hainan Province of China (No. 319MS082) and the Central Public Interest Scientific Institution Basal Research Fund (No. 1630032016006).

## Author contributions

L.X. and Z.Y.L. designed the research; Z.Y.L., J.B.W., Y.L.J, B.L.H., Y.L.F., and X.B.W. performed the experiments, L.X., Z.Y.L. and J.B.W. analysed the data; Q.Q.Y. and C.Y.M. supplied the materials; J.B.W. and Z.Y.L. wrote the manuscript, and L.X., X.B.Z. and G.H.Z. revised the manuscript. All authors critically read and approved the final version of the manuscript.

## Competing interests

The authors declare no competing interests.

**Additional information**

