## [Peer Review File · Communications Biology]

Reviewers' comments:

Reviewer #1 (Remarks to the Author):

The manuscript titled "The genome of *Aechmea fasciata* provides insight into the evolution of tank habit and ethylene-induced flowering" aims to reveal the genomic basis of tank habit formation and ethylene-induced flowering of *Aechmea fasciata*, one of the most popular ornamental bromeliads bearing a water-impounding tank, which is a key innovation that promoted the diversity of this plant family. The authors sequenced the genome of *A. fasciata* and assembled 352 Mb of sequences into 24 chromosomes. Comparative genomic analysis showed that its chromosomes experienced at least two fissions and two fusions from the ancestral genome of *A. fasciata* and *Ananas comosus* F153. The gibberellin receptor gene *GID1C*-like was duplicated during a segmental duplication event and may promote rosette expansion, allowing water impounding tank formation. They consider that these data will facilitate further research on the genomic basis of key functional traits and serve as an important resource for studying the evolution of bromeliads and mechanisms underlying their flowering time regulation.

The manuscript provides some interesting new data about this topic, as to date few investigations have been conducted on the molecular mechanism underlying ethylene-induced flowering in bromeliads, while the genomic basis of functional traits, such as CAM photosynthesis and tank habit, is poorly understood. In general, the study design, analyses and use of statistics seem to be well developed; however, the whole manuscript still requires a careful revision of its writing style, and the references need to be checked considering the journals guidelines. Furthermore, there are several points mainly within the introduction and discussion that remain unclear or incomplete, which should be revised and explained/discussed in more detail (see comments made within the document). For example, it should be described how and why the mentioned ecological and morphological adaptations can be considered as key innovations for bromeliad radiation. Also, in which way the genome sequences of *A. fasciata* will facilitate further research on bromeliad evolution and which key roles the expanded gene families may play. Furthermore, they should give some examples of genera of epiphytic bromelioids or tank-less bromeliads.

Reviewer #2 (Remarks to the Author):

The genome of *Aechmea fasciata* provides insight into the evolution of tank habit and ethylene-induced flowering claims to have found the key effectors for the tank habit in *A. fasciata* being associated with duplication of *GID1C*-like genes and contraction of *DELLA*-like genes. In addition, they identified the *DELLA*-like genes are potential effectors that functions as a flowering repressor and plays essential roles in ethylene-delayed flowering. In addition to the genomic work and citation of literature, the authors were able to test their hypothesis using transgenic lines of the *A. fasciata* expressing the flowering genes. Although it is probably beyond the scope of this manuscript, it would be interesting to see if how externally modified GA applications or inhibition of GA levels through transgene or plant growth regulator application will interact with ethylene induced flowering. This work is important in understanding the intricacies of plant hormone interactions for plant adaptation and evolution. Minor editing suggestions are provided in the file.

Reviewer #3 (Remarks to the Author):

The manuscript by Li et., al. presents a chromosomal-scale genome assembly of *Aechmea fasciata*. Comparative genomics between *Aechmea fasciata* and its close relatives pineapple varieties F153 and CB5 showed evolutionary history of ancestral genomes. Integrated analysis using genomics and transcriptomics revealed that duplicated *GID1C*-like gene may contribute to rosette expansion through affecting GA signalling. The authors also performed Chip-seq analysis to validate that *AfFTL2*, an

ethylene-induced flowering gene, could be regulated by AFEIL1 transcription factor. Overall, this manuscript is a typical genome analysis research and tries to address several species-specific questions based on a high-quality genome assembly and multiple-omics data. Although this study provides useful genomics resources, I still have some concerns.

1. *Aechmea fasciata* contains 18,644 protein-coding genes, much fewer than that in pineapple (~27 k in F153 and ~29k in CB5). Is there any reason to explain why the two closely related species have such a large difference in gene number. Did the authors check the completeness of these annotated genes?

2. The authors stated that *GID1C*-like maybe the key gene to promote rosette expansion. However, I did not find enough evidences to support the conclusion.

3. It is nice to see that the authors performed comparative analysis between *Aechmea fasciata* and two pineapple varieties (var. *Comosus* and var. *bracteatus*). However, it seems that they did not notice there were two F153 assembly versions released. The first version published in 2015 contains chimeric assembly errors, which have been corrected in the second version published in 2019. If the first version was used for the comparative analysis, the ancestral genome construction may need to be re-analyzed.

4. F153 and CB5 are commercial names, representing two different pineapple varieties, var. *comosus* and var. *bracteatus*. I strongly suggest to use the Latin names instead of commercial names in a scientific manuscript.

5. Citations are missing in a couple of statements, for instance, Line 130 MARCA2, Line 149 DupGeneFinder, Line 159 "when the core Bromeliaceae appeared in the Atlantic rainfall forest" and Line 160 cafe.

Reviewer #4 (Remarks to the Author):

In this manuscript author sequenced and assembled the genome of one horticultural plant species *Aechmea fasciata*. They compare this genome with two varieties of *Ananas comosus*, another bromeliad species. They also discussed the evolution of some genes (gibberellin receptor gene *GID1C*-like and ethylene-induced flowering) likely involved in the promoting rosette expansion, allowing water impounding tank formation, an ecological and key innovation trait in Bromeliaceae.

The genome sequencing was based on Nanopore, Pacbio, Illumina and Hi-C sequencing to achieve a high level of completeness and contiguity of the genome. In my point of view, this are the critical part of the study. First, they reported a BUSCO completeness of only 92.2% of the BUSCO orthologues, which suggest there must be important amount of genes not reported. It maybe because PacBio reads were in average length of 9,377.4 bp which is too small for a plant, now PacBio HiFi reads are in average 20-15kbp. These longer reads will help to improve the completeness of the genome.

Also, the genome size assemble is far too small than the expected based on the other two bromeliads with reference genomes available. They discussed this unexpected results based on lost of LTR and gene sequences, which were not convincing considering the other results. For instance, *Aechmea fasciata* has n=25 chromosome (Marchant et al 1967 doi.org/10.2307/4108461; Paule et al 2020 - Front Plant Sci. doi: 10.3389/fpls.2020.01295), but in this study authors assembled only 24 chromosomes, and do not discussed this discrepancy with the former chromosome number reported. Since, chromosome number do not change much in Bromeliaceae with almost all species being n=25 chromosomes, I do not see a biological reason to accept *Aechmea fasciata* is n=24 chromosomes, and together with the small genome size reported, I wonder whether is this a methodological issue here. One easy way to solve that is to present the chromosome number by conventional cytogenetics of the plant sequenced in this study and include longer reads as those from PacBio HiFi.

Another major issue in the manuscript is the introduction section. I found several problems, then I suggest a deep review in the biological section of the introduction, see my comments bellow: Line 57-58. There is a constant change in number of genera and species in the family. I suggest not include the exact number, you may indicate > 75 genera and >3500sp. The references used here are not corrected for this citation, please see Gouda EJ, Butcher D, Gouda K. cont. updated. Encyclopaedia of bromeliads, Version 4. University Utrecht Botanic Gardens. Available at:

<http://bromeliad.nl/encyclopedia/>

Line 58. Bromeliads from core Bromelioid clade one clade of the Bromelioideae sub-family, fill in the descriptions made, but pay attention those represent less than 1/5 of the family.

in lines 62-63 you reported that Tillandsioideae being diploid with $2n=48$. Which is not corrected following Gitai et al 2014.

Most of Tillandsioideae (*Vriesea*, *Alcantarea*, *Catopsis*, *Racinea*, *Guzmania*, *Tillandsia*, etc) are $2n=50$. The original sentence in Gitai et al 2014 is: "Most species were diploid with $2n = 50$ in Bromelioideae, Puyoideae and Pitcairnioideae, followed by $2n = 48$ observed mainly in Tillandsioideae." It means, that $2n=48$, although rare is mainly, almost only found in Tillandsioideae, not that all Tillandsioideae has $2n=48$...

66-67. this information is also incorrect. Please see Givnish et al 2014 and Silvestro et al 2014.

Lines 73-89. The information on this paragraph is superficial, meaning that one who are not familiar can not understand what all these genes and pathways are responsible for. On the other hand these informations are not connect with the rest of the introduction.

1. Please revise the text of the manuscript. The text is currently very difficult to follow due to grammatical errors, certain points and conclusions from the existing data are overstated and references are missing or incorrect. Please correct accordingly.

Response: Thank you for the above suggestion. we carefully revised the text. And the manuscript was edited by AJE (American Journal Experts) for proper English language, grammar, punctuation, spelling, and overall style.

2. The authors should consider if they can address the point of reviewer #2 regarding how externally modified GA applications or inhibition of GA levels through transgene or plant growth regulator application will interact with ethylene induced flowering.

Response: Thank you for the above suggestion. In our unpublished study, we demonstrated that ethylene treatment increased the protein level of *AfSLN1* and decreased the GA levels in *A. fasciata*, but had little effect on ethylene induced flowering. GA treatment cannot induce flowering of *A. fasciata*, which was proved in our previous experiments.

3. Please address reviewer #3's comment regarding the low number of protein-coding genes in *Aechmea fasciata* when compared to pineapple.

Response: Thank you for the above suggestion. we annotated the protein-coding genes again, using Trinity produced transcripts as supplementary EST evidence for Maker. As a result, we got 20,953 protein-coding genes, an increase of 2,309 protein-coding genes compared to previous version. Analysis of gene family and gene duplication were processed again, and contents in Fig. 3 and table S5 were modified accordingly.

4. Provide additional support or explanation regarding how *GID1C*-like maybe the key gene to promote rosette expansion.

Response: Thank you for the above suggestion. We have added more explanation and references in lines 213-217 ("Overexpression of *OsGID1C* resulted in a GA hypersensitivity phenotype in transgenic rice⁴⁶. Overexpression of *GID1C-like* in *Arabidopsis* resulted in expansion of rosettes^{47,48}. The tandem duplication of the *GID1C-like* gene increased its expression level and may improve GA sensitivity and promote rosette expansion in *A. fasciata*, which allows the formation of tanks with closely overlapping leaves.").

5. Ensure as noted by Reviewer #3 that the second version of the F153 genome published in 2019 was used for analysis.

Response: Thank you for the above suggestion. The dot plot between *A. comosus* and *A. fasciata*, and genome ancestry analysis were conducted with the genome version published in 2019.

6. Ensure to use Latin names when referring to pineapple.

Response: Thank you for the above suggestion. We replaced F153, cb5 with *Ananas comosus* and *Ananas bracteatus*.

Reviewer #1 (Remarks to the Author)

1. The manuscript titled "The genome of *Aechmea fasciata* provides insight into the evolution of tank habit and ethylene-induced flowering" aims to reveal the genomic

basis of tank habit formation and ethylene-induced flowering of *Aechmea fasciata*, one of the most popular ornamental bromeliads bearing a water-impounding tank, which is a key innovation that promoted the diversity of this plant family. The authors sequenced the genome of *A. fasciata* and assembled 352 Mb of sequences into 24 chromosomes. Comparative genomic analysis showed that its chromosomes experienced at least two fissions and two fusions from the ancestral genome of *A. fasciata* and *Ananas comosus* F153. The gibberellin receptor gene GID1C-like was duplicated during a segmental duplication event and may promote rosette expansion, allowing water impounding tank formation. They consider that these data will facilitate further research on the genomic basis of key functional traits and serve as an important resource for studying the evolution of bromeliads and mechanisms underlying their flowering time regulation.

In general, the study design, analyses and use of statistics seem to be well developed; however, the whole manuscript still requires a careful revision of its writing style, and the references need to be checked considering the journals guidelines. Furthermore, there are several points mainly within the introduction and discussion that remain unclear or incomplete, which should be revised and explained/discussed in more detail (see comments made within the document). For example, it should be described how and why the mentioned ecological and morphological adaptations can be considered as key innovations for bromeliad radiation. Also, in which way the genome sequences of *A. fasciata* will facilitate further research on bromeliad evolution and which key roles the expanded gene families may play. Furthermore, they should give some examples of genera of epiphytic bromelioids or tank-less bromeliads.

Response: Thank you for the above suggestion. we carefully revised the manuscript and references. In Line:71-75, we explained why CAM and tank-habit are key innovations for bromeliad radiation and added examples of genera of epiphytic bromelioids or tank-less bromeliads. The roles of expanded family may play were discussed in lines 300-318 and lines 347-360.

Reviewer #2 (Remarks to the Author):

1. The genome of *Aechmea fasciata* provides insight into the evolution of tank habit and ethylene-induced flowering claims to have found the key effectors for the tank habit in *A. fasciata* being associated with duplication of GID1C-like genes and contraction of DELLA-like genes. In addition, they identified the DELLA-like genes are potential effectors that functions as a flowering repressor and plays essential roles in ethylene-delayed flowering. In addition to the genomic work and citation of literature, the authors were able to test their hypothesis using transgenic lines of the *A. fasciata* expressing the flowering genes. Although it is probably beyond the scope of this manuscript, it would be interesting to see if how externally modified GA applications or inhibition of GA levels through transgene or plant growth regulator application will interact with ethylene induced flowering. This work is important in understanding the intricacies of plant hormone interactions for plant adaptation and evolution. Minor editing suggestions are provided in the file.

Response: Thank you for the above suggestion. In previous studies of pineapple, ethylene treatment decreased GA levels, similar to the reports in *Arabidopsis*. Furthermore, we quantified the protein levels of *AfSLN1*, which increased in response to ethylene treatment (unpublished). But the homologues of flowering repressor *SVP* and *ELF3* were not increased by *DELLA* in *A. fasciata*, unlike in *Arabidopsis*. Therefore, we supposed that decreased GA levels have little effects on ethylene induced flowering, may partially due to the specific motifs of *DELLAs*. Other more, GA treatment cannot induce flowering of *A. fasciata*, which was proved in our previous experiments.

Reviewer #3 (Remarks to the Author):

The manuscript by Li et., al. presents a chromosomal-scale genome assembly of *Aechmea fasciata*. Comparative genomics between *Aechmea fasciata* and its close relatives pineapple varieties F153 and CB5 showed evolutionary history of ancestral genomes. Integrated analysis using genomics and transcriptomics revealed that duplicated *GID1C*-like gene may contribute to rosette expansion through affecting GA signalling. The authors also performed Chip-seq analysis to validate that *AfFTL2*, an ethylene-induced flowering gene, could be regulated by *AfEIL1* transcription factor. Overall, this manuscript is a typical genome analysis research and tries to address several species-specific questions based on a high-quality genome assembly and multiple-omics data. Although this study provides useful genomics resources, I still have some concerns.

1. *Aechmea fasciata* contains 18,644 protein-coding genes, much fewer than that in pineapple (~27 k in F153 and ~29k in CB5). Is there is any reason to explain why the two closely related species have such a large difference in gene number. Did the authors check the completeness of these annotated genes?

Response: Thank you for the above suggestion. To improve the annotation of *A. fasciata*, we added Trinity produced transcripts for EST evidences for MAKER. And then was revised by PASA pipeline. As a result, we got 20,953 protein-coding genes, an increase of 2,309 protein-coding genes compared to previous version. The number of protein-coding genes close to the 22,166 of *A. comosus*, which were annotated by NCBI pipeline. Analysis of gene family and gene duplication were processed again, and contents in Fig. 3 and table S5 were modified accordingly. Difference between methods and parameters may lead to the difference of gene numbers between *A. fasciata* and *A. comosus*. And the gene-duplication and gene family contraction and expansion analysis were processed again. According to the results, the fewer gene family expansion and more gene family contraction than *A. comosus*, may be the main reason of fewer genes than *A. comosus*.

2. The authors stated that *GID1C*-like maybe the key gene to promote rosette expansion. However, I did not find enough evidences to support the conclusion.

Response: Thank you for the above suggestion. We revised corresponding discussion in lines 213-217 (“Overexpression of *OsGID1C* resulted in a GA hypersensitivity phenotype in transgenic rice⁴⁶. Overexpression of *GID1C-like* in *Arabidopsis* resulted in expansion of rosettes^{47,48}. The tandem duplication of the *GID1C-like* gene increased its expression level and may improve GA sensitivity and promote rosette expansion in *A. fasciata*, which allows the formation of tanks with closely overlapping leaves.”).

3. It is nice to see that the authors performed comparative analysis between *Aechmea*

fasciata and two pineapple varieties (var. *Comosus* and var. *bracteatus*). However, it seems that they did not notice there were two F153 assembly versions released. The first version published in 2015 contains chimeric assembly errors, which have been corrected in the second version published in 2019. If the first version was used for the comparative analysis, the ancestral genome construction may need to be re-analyzed.

Response: Thank you for the above suggestion. The dot plot *A. comosus* and *A. fasciata*, and genome ancestry analysis were conducted with the genome version published in 2019.

4. F153 and CB5 are commercial names, representing two different pineapple varieties, var. *comosus* and var. *bracteatus*. I strongly suggest to use the Latin names instead of commercial names in a scientific manuscript.

Response: Thank you for the above suggestion. we replaced F153 and CB5 with their Latin names in the manuscript.

5. Citations are missing in a couple of statements, for instance, Line 130 MARCA2, Line 149 DupGeneFinder, Line 159 “when the core Bromeliaceae appeared in the Atlantic rainfall forest” and Line 160 *cafe*.

Response: Thank you for the above suggestion. We added corresponding citations and references.

Reviewer #4 (Remarks to the Author):

In this manuscript author sequenced and assembled the genome of one horticultural plant species *Aechmea fasciata*. They compare this genome with two varieties of *Ananas comosus*, another bromeliad species. They also discussed the evolution of some genes (gibberellin receptor gene *GID1C*-like and ethylene-induced flowering) likely involved in the promoting rosette expansion, allowing water impounding tank formation, an ecological and key innovation trait in Bromeliaceae.

1. The genome sequencing was based on Nanopore, Pacbio, Illumina and Hi-C sequencing to achieve a high level of completeness and contiguity of the genome. In my point of view, this are the critical part of the study. First, they reported a BUSCO completeness of only 92.2% of the BUSCO orthologues, which suggest there must be important amount of genes not reported. It maybe because PacBio reads were in average length of 9,377.4 bp which is too small for a plant, now PacBio HiFi reads are in average 20-15kbp. These longer reads will help to improve the completeness of the genome.

Also, the genome size assemble is far too small than the expected based on the other two bromeliads with reference genomes available. They discussed these unexpected results based on loss of LTR and gene sequences, which were not convincing considering the other results. For instance, *Aechmea fasciata* has $n=25$ chromosome (Marchant et al 1967 doi.org/10.2307/4108461; Paule et al 2020 - Front Plant Sci. doi: 10.3389/fpls.2020.01295), but in this study authors assembled only 24 chromosomes, and do not discussed this discrepancy with the former chromosome number reported. Since, chromosome number do not change much in Bromeliaceae with almost all species being $n=25$ chromosomes, I do not see a biological reason to accept *Aechmea fasciata* is $n=24$ chromosomes, and together with the small genome size reported, I wonder whether is this a methodological issue here. One easy way to solve that is to present the

chromosome number by conventional cytogenetics of the plant sequenced in this study and include longer reads as those from PacBio HiFi.

Response: Thank you for the above suggestion. In this study, Nanopore reads were used for genome assembly and PacBio reads and Illumina short reads were used for correction of genome assembly. We firstly assembled genome sequences using PacBio reads, and obtained 24 chromosomes with 300 Mb sequences after Hi-c scaffolding using 3d-DNA and Juicebox. To improve the completeness of genome assembly, we then used Nanopore reads and obtained a genome assembly with 352 Mb after Hi-c scaffolding using ALLHiC. Using BUSCO to assess completeness of genome with *viridiplantae_odb10*, results show that 97.0%, 96.7%, 87.0% BUSCO orthologues were included in *A. comosus*, *A. fasciata* and *A. bracteatus*, respectively. And the genome assembly size of 352 Mb was close to the estimated genome size of 359 Mb, which may be smaller than estimated size of previous studies due to methodological difference. And the assembly size of *A. fasciata* was close to the genome assembly size of *A. comosus*. Therefore, we supposed that the completeness of *A. fasciata* genome assembly was in a high level.

Using Hi-c scaffolding, we obtained 24 pseudo-chromosomes. To validate this, we counted chromosome numbers by root tip squash method. The result showed that *A. fasciata* had 48 chromosomes normally, which was added as Fig. S11. Because two of the chromosomes (chromosome 1) have nearly twofold length than other chromosomes and were fragile, we also found 50 chromosomes in some cells. Therefore, the reported $2n=50$ for *A. fasciata* in previous studies may be due to the breakage of chromosome 1 near the centromere. We added the corresponding discussions in lines 279-282 ("Due to the chromosome fusions of ancestry chromosomes, *A. fasciata* has 48 ($2n=48$) chromosomes and was validated by the root tip squash method (Fig. S11). We also observed a karyotype with $2n=50$ due to breakage of chromosome 1 near the centromere, which may be the reason why previous reports described a karyotype with $2n = 50$ chromosomes⁵⁷.").

2. Another major issue in the manuscript is the introduction section. I found several problems, then I suggest a deep review in the biological section of the introduction, see my comments below:

Line 57-58. There is a constant change in number of genera and species in the family. I suggest not include the exact number, you may indicate > 75 genera and >3500sp. The references used here are not corrected for this citation, please see Gouda EJ, Butcher D, Gouda K. cont. updated. Encyclopaedia of bromeliads, Version 4. University Utrecht Botanic Gardens. Available at: <http://bromeliad.nl/encyclopedia/> Line 58. Bromeliads from core Bromelioid clade one clade of the Bromelioideae sub-family, fill in the descriptions made, but pay attention those represent less than 1/5 of the family. In lines 62-63 you reported that Tillandsioideae being diploid with $2n=48$. Which is not corrected following Gitai et al 2014. Most of Tillandsioideae (*Vriesea*, *Alcantarea*, *Catopsis*, *Racinea*, *Guzmania*, *Tillandsia*, etc) are $2n=50$. The original sentence in Gitai et al 2014 is: "Most species were diploid with $2n = 50$ in Bromelioideae, Puyoideae and Pitcairnioideae,

followed by $2n = 48$ observed mainly in Tillandsioideae." It means, that $2n=48$, although rare is mainly, almost only found in Tillandsioideae, not that all Tillandsioideae has $2n=48$... 66-67. this information is also incorrect. Please see Givnish et al 2014 and Silvestro et al 2014. Lines 73-89. The information on this paragraph is superficial, meaning that one who are not familiar cannot understand what all these genes and pathways are responsible for. On the other hand, these informations are not connect with the rest of the introduction.

Response: Thank you for the above suggestion. We revised the introduction section, especially the parts related to flowering. And the incorrect quotes were revised, such as in lines 57-58 and 64 ("Bromeliaceae contains more than 75 genera and more than 3500 species and is the largest family of flowering plants found in the neotropics¹⁻³", "One of the most diverse clades from Bromeliaceae, core Bromelioids often use CAM photosynthesis..." and "with most species of Bromelioideae, Puyoideae, Tillandsioideae and Pitcairnioideae being diploid at $2n=50$ and some Tillandsioideae being diploid at $2n=48$ ^{7,8}"). As ethylene induced flowering is a remarkable feature of Bromeliaceae species, which is exploited worldwide to promote flowering synchronization. However, this is an exceptional case, and ethylene generally inhibits flowering in many plant species. Therefore, the molecular mechanism underlying ethylene-induced flowering are research hotspots in bromeliads. And as an important part of this study, we carefully revised the paragraphs related to flowering mechanism.

Reviewers' comments:

Reviewer #2 (Remarks to the Author):

The authors of "The genome of *Aechmea fasciata* provides insight into the evolution of tank epiphytic habits and ethylene-induced flowering" have revised the manuscript and in my opinion, answered the reviewer's comments. The authors have conducted extensive work to test hypotheses on the involvement of specific genes in flowering. The manuscript is much easier to read with the corrections. Although not complete, the information presented by this manuscript does present interesting and important insight into dissecting the ethylene-induced flowering of *Ananas* species.

Reviewer #3 (Remarks to the Author):

The authors have addressed most of my concerns. However, there are still two issues that need to be fixed.

1. I am not the expert on pineapple. But as far as I know, *Ananas comosus* and *Ananas bracteatus* have been classified into one single species. That's why I said that they are two varieties. And the Latin names are supposed to be *Ananas comosus* var. *comosus* and *Ananas comosus* var. *bracteatus*.
2. In my first question, I raised concerns on the low number of protein-coding genes. I understand that different annotation approaches may lead to quite a different number of gene models. But they should at least provide a simple assessment of their annotation. I asked a BUSCO analysis on the annotation before, however, I did not see the report in this revised manuscript.

Reviewer #5 (Remarks to the Author):

The authors sequenced and analyzed the genome of the commonly cultivated bromeliad *Aechmea fasciata*. They did comparative genomic analyses with pineapple (*Ananas comosus*) and the *bracteatus* variety of pineapple, as well as specific gene family analyses.

While the genome assembly is of use to the angiosperm/monocot community, there are a number of issues with the manuscript. First and foremost is the numerous grammatical, spelling, and editing issues. There are too many to list. Citations and capitalizations are missing for nearly every program used.

The gene family portion of the introduction is much too detailed. Also, Bromeliaceae is not the most species rich family of angiosperms in the neotropics (see Orchidaceae).

It is not clear in the manuscript that *A. bracteatus* is just a variety of *Ananas comosus*. That also makes me question their divergence times, as they state the divergence of the two *Ananas* taxa at 3 million years. *Bracteatus* is "anciently cultivated" but the study of that genome (Chen et al., 2019) places the initial cultivation of pineapple ~6,000 years.

The many gene family analyses are very disorganized and unclear in the underlying reasons for investigation and overall messages.

Overall, this manuscript needs an overhaul in terms of editing and organization.

Reviewers' comments:

Reviewer #2 (Remarks to the Author):

The authors of "The genome of *Aechmea fasciata* provides insight into the evolution of tank epiphytic habits and ethylene-induced flowering" have revised the manuscript and in my opinion, answered the reviewer's comments. The authors have conducted extensive work to test hypotheses on the involvement of specific genes in flowering. The manuscript is much easier to read with the corrections. Although not complete, the information presented by this manuscript does present interesting and important insight into dissecting the ethylene-induced flowering of *Ananas* species.

Response: Thank you for the above suggestion.

Reviewer #3 (Remarks to the Author):

The authors have addressed most of my concerns. However, there are still two issues that need to be fixed.

1. I am not the expert on pineapple. But as far as I know, *Ananas comosus* and *Ananas bracteatus* have been classified into one single species. That's why I said that they are two varieties. And the Latin names are supposed to be *Ananas comosus* var. *comosus* and *Ananas comosus* var. *bracteatus*.

Response: Thank you for the above suggestion. We renamed *Ananas comosus* and *Ananas bracteatus* with *Ananas comosus* var. *bracteatus* and *Ananas comosus* var. *bracteatus* throughout the entire manuscript.

2. In my first question, I raised concerns on the low number of protein-coding genes. I understand that different annotation approaches may lead to quite a different number of gene models. But they should at least provide a simple assessment of their annotation. I asked a BUSCO analysis on the annotation before, however, I did not see the report in this revised manuscript.

Response: Thank you for the above suggestion. We assessed the protein-coding genes (20,953) using BUSCO. The results showed 76.2% of BUSCO orthologues were completed. To improve the completeness of annotated protein-coding genes, we used BRAKER predicted gene models as input for MAKER pipeline. As a result, we got 26,126 genes and BUSCO assessment showed that 93.4% of BUSCO orthologues were completed. The assessment table were added as table S3. And we updated BUSCO to v5.2.2 and used the viridiplantae_odb10 dataset.

Reviewer #5 (Remarks to the Author):

The authors sequenced and analyzed the genome of the commonly cultivated bromeliad *Aechmea fasciata*. They did comparative genomic analyses with pineapple (*Ananas comosus*) and the bracteatus variety of pineapple, as well as specific gene family analyses.

While the genome assembly is of use to the angiosperm/monocot community, there are a number of issues with the manuscript. First and foremost is the numerous grammatical, spelling, and editing issues. There are too many to list. Citations and capitalizations are missing for nearly every program used.

Response: Thank you for the above suggestion. We carefully revised the manuscript. And the manuscript was edited by AJE (American Journal Experts) again.

The gene family portion of the introduction is much too detailed. Also, Bromeliaceae is not the most species rich family of angiosperms in the neotropics (see Orchidaceae).

Response: Thank you for the above suggestion. We carefully revised the manuscript. We rewritten the sentence as “is one of the largest families of flowering plants found in the neotropics” in L57-L58. We deleted the sentences about the mechanism of flowering promotion by florigen in L83-L89.

It is not clear in the manuscript that *A. bracteatus* is just a variety of *Ananas comosus*. That also makes me question their divergence times, as they state the divergence of the two *Ananas* taxa at 3 million years. *Bracteatus* is “anciently cultivated” but the study of that genome (Chen et al., 2019) places the initial cultivation of pineapple ~6,000 years.

Response: Thank you for the above suggestion. We added the discussions as “Although *A. comosus* var. *bracteatus* is a variety of pineapple, the introgression of *A. macrodontes* genes into *A. comosus* led *A. comosus* var. *bracteatus* and *A. comosus* var. *comosus* to have a certain evolutionary distance.” in L274-L278. We deleted the descriptions about divergence time between *A. comosus* var. *bracteatus* and *A. comosus* var. *comosus*.

The many gene family analyses are very disorganized and unclear in the underlying reasons for investigation and overall messages.

Response: Thank you for the above suggestion. We deleted the sentences about genes involved in disease resistance in L203-L212 and L369-L373. Added titles in L183 and L202.

Overall, this manuscript needs an overhaul in terms of editing and organization.

Response: Thank you for the above suggestion. We carefully revised the manuscript.

REVIEWERS' COMMENTS:

Reviewer #5 (Remarks to the Author):

This is a revised manuscript of the *Aechmea fasciata* genome. While I still think the resources from this study are important for the flowering plant/monocot community, there are still a number of organizational/writing errors that make this manuscript very difficult to read.

The Introduction is quite scattered in its message, alternating between Bromeliaceae diversity to adaptations to very specific genetic pathways to comparative genomics. This section needs to be simplified and stream-lined to give the reader a clear thesis for the study.

Line 56: silver vase, not verse

Line 66: bromeliads should be lower case

Line 99: you never spell out *Ananas comosus*, nor that it is pineapple

The Results can also be simplified by simply stating the results/stats rather and moving the methods/programs to the Methods section. For example, Line 127-128 can simply state "We identified 1,474 pseudogenes and 1,239 transcription factors."

Line 124: marker should be capitalized and needs and reference. Is that Maker?

Line 144: ancestry should be ancestral

Line 158: anything interesting to note from these gene family sizes? If not, maybe move to suppl.

Line 165: all gene families in *Aechmea* were decreased by more than 1?

Line 174: expand on the shared WGD event. When was it? Is that event already known/published?

Line 216: no need for precise coordinates

Line 258: ChIP-seq

Much like the Introduction, the Discussion is largely disorganized and lacks a flow. It jumps from topic to topic, sometimes describing results that were never noted in the Results section. It also lacks a clear concluding paragraph.

Line 278: "have a certain evolutionary distance" is very obscure

Line 282: specific translations?

Line 292: what previous report?

Line 292: Why "despite"?

Line 296: ancestral and italicize *A. fasciata*

Line 311: I don't think I understand the LIMYB sentence, nor was it discussed in the results

Line 314: "large" genes?

Line 319: what's episodic? The loss of WGD gene pair loss? Why would that be episodic?

Line 324: poor sentence structure

Line 327: degeneration?

Line 362: were these defense genes ever mentioned in results?

REVIEWERS' COMMENTS:

Reviewer #5 (Remarks to the Author):

This is a revised manuscript of the *Aechmea fasciata* genome. While I still think the resources from this study are important for the flowering plant/monocot community, there are still a number of organizational/writing errors that make this manuscript very difficult to read.

The Introduction is quite scattered in its message, alternating between Bromeliaceae diversity to adaptations to very specific genetic pathways to comparative genomics. This section needs to be simplified and stream-lined to give the reader a clear thesis for the study.

Response: Thank you for the above suggestion. We carefully revised the introduction section. The section about flowering genetic pathways was simplified.

Line 56: silver vase, not verse

Response: Thank you for the above suggestion. We revised this writing error.

Line 66: bromeliads should be lower case

Response: Thank you for the above suggestion. We revised this writing error.

Line 99: you never spell out *Ananas comosus*, nor that it is pineapple

Response: Thank you for the above suggestion. We revised this writing error.

The Results can also be simplified by simply stating the results/stats rather and moving the methods/programs to the Methods section. For example, Line 127-128 can simply state "We identified 1,474 pseudogenes and 1,239 transcription factors."

Response: Thank you for the above suggestion. We simplified the result section by moving some programs/methods into Methods section such in L133, L134, L147 and L164.

Line 124: marker should be capitalized and needs and reference. Is that Maker?

Response: Thank you for the above suggestion. We revised this writing error and added reference.

Line 144: ancestry should be ancestral

Response: Thank you for the above suggestion. We revised this writing error.

Line 158: anything interesting to note from these gene family sizes? If not, maybe move to suppl.

Response: Thank you for the above suggestion. There is much interest in gene family expansion and contraction, as even the gain or loss of single genes have been implicated in adaptive divergence between species.

Line 165: all gene families in *Aechmea* were decreased by more than 1?

Response: Thank you for the above suggestion. We revised the sentence as “the number of gene families of size more than 1 was decreased significantly in *A. fasciata*” in L170-L171.

Line 174: expand on the shared WGD event. When was it? Is that event already known/published?

Response: Thank you for the above suggestion. We revised the sentence as “There were similar WGD peaks in the three genomes, corresponding to WGD event occurred 100–120 million years ago” in L178-L179.

Line 216: no need for precise coordinates

Response: Thank you for the above suggestion. We revised the sentence as “The gibberellin (GA) receptor gene *GID1C* was duplicated in *A. fasciata* due to segmental tandem duplication in a region of chr05, which is a conserved syntenic region with that of chr05 of *A. comosus* var. *comosus* and chr11 of *A. comosus* var. *bracteatus*” in L207-L209.

Line 258: ChIP-seq

Response: Thank you for the above suggestion. We revised this writing error.

Much like the Introduction, the Discussion is largely disorganized and lacks a flow. It jumps from topic to topic, sometimes describing results that were never noted in the Results section. It also lacks a clear concluding paragraph.

Response: Thank you for the above suggestion. We carefully revised the discussion section.

Line 278: “have a certain evolutionary distance” is very obscure

Response: Thank you for the above suggestion. We revised the sentence as “the introgression of *A. macrodontes* genes into *A. comosus* led to genetic differentiation between *A. comosus* var. *bracteatus* and *A. comosus* var. *comosus*” in L283-L285.

Line 282: specific translations?

Response: Thank you for the above suggestion. We revised the sentence as “A.

comosus var. *bracteatus* experienced specific translocations on chr23" in L288.

Line 292: what previous report?

Response: Thank you for the above suggestion. We revised the sentence as "which is close to the time of the tank-forming core-Bromelioideae arising in the Atlantic Forest" in 282-283.

Line 292: Why "despite"?

Response: Thank you for the above suggestion. We revised the sentence as "In addition to multiple translocations, inversions and shifts of chromosomes" in L288-L289.

Line 296: ancestral and italicize *A. fasciata*

Response: Thank you for the above suggestion. We revised this sentence.

Line 311: I don't think I understand the LIMYB sentence, nor was it discussed in the results

Response: Thank you for the above suggestion. We deleted this sentence.

Line 314: "large" genes?

Response: Thank you for the above suggestion. We revised the sentence as "The loss of numerous genes".

Line 319: what's episodic? The loss of WGD gene pair loss? Why would that be episodic?

Response: Thank you for the above suggestion. We revised the sentence as "It is supposed that WGD pairs were lost in the subsequent millions of years" in L313-L314. The WGD events are episodic.

Line 324: poor sentence structure

Response: Thank you for the above suggestion. We revised the sentence as "With the formation of tank habits, bromeliads invaded the treetops of rainforests. Especially in Atlantic forests of Brazil, they presented with extraordinary biodiversity." in L319-L320.

Line 327: degeneration?

Response: Thank you for the above suggestion. We revised the sentence as "leading to their degeneration into anchorage".

Line 362: were these defense genes ever mentioned in results?

Response: Thank you for the above suggestion. These genes were mentioned in L189-L205.